

# Constraining the SMEFT with Bayesian reweighting

**Samuel van Beek,[1] Emanuele R. Nocera,[1*] Juan Rojo[1,2] and Emma Slade[3]**

**1** Nikhef Theory Group, Science Park 105, 1098 XG Amsterdam, The Netherlands.
**2** Department of Physics and Astronomy, Vrije Universiteit, 1081 HV Amsterdam.
**3** Rudolf Peierls Centre for Theoretical Physics, University of Oxford,
Clarendon Laboratory, Parks Road, Oxford OX1 3PU, United Kingdom

* e.nocera@nikhef.nl

## Abstract

We illustrate how Bayesian reweighting can be used to incorporate the constraints provided by new measurements into a global Monte Carlo analysis of the Standard Model Effective Field Theory (SMEFT). This method, extensively applied to study the impact of new data on the parton distribution functions of the proton, is here validated by means of our recent SMEFiT analysis of the top quark sector. We show how, under well-defined conditions and for the SMEFT operators directly sensitive to the new data, the reweighting procedure is equivalent to a corresponding new fit. We quantify the amount of information added to the SMEFT parameter space by means of the Shannon entropy and of the Kolmogorov-Smirnov statistic. We investigate the dependence of our results upon the choice of alternative expressions of the weights.

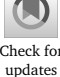
# 1 Introduction

A powerful framework to parametrise and constrain potential deviations from Standard Model (SM) predictions in a model-independent way is provided by the SM Effective Field Theory (SMEFT) [1–3], see [4] for a recent review. In the SMEFT, effects of beyond the SM (BSM) dynamics at high scales $E \simeq \Lambda$ are parametrised, for $E \ll \Lambda$, in terms of higher-dimensional operators built up from the SM fields and symmetries. This approach is fully general, as one can construct complete bases of independent operators, at any given mass dimension, that can be systematically matched to ultraviolet-complete theories.

Analysing experimental data in the SMEFT framework, however, is far from straightforward because of the large dimensionality of the underlying parameter space. For instance, without flavour assumptions, one needs to deal with $N_{\mathrm{op}} = 2499$ independent operators corresponding to three fermion generations. Because of this challenge, the complexity and breadth of SMEFT analyses, in particular of LHC data, has been restricted to a subset of higher-dimensional operators so far, typically clustered in sectors that are assumed to be independent from each other [5–22].

More recently, some of us have developed a novel approach to efficiently explore the parameter space in a global analysis of the SMEFT: the SMEFiT framework [23]. This approach is inspired by the NNPDF methodology [24–28] for the determination of the parton distribution functions (PDFs) of the proton [29–31]. The SMEFiT methodology realises a Monte Carlo representation of the probability distribution in the space of the SMEFT parameters, whereby each parameter is associated to a statistical ensemble of equally probable replicas. Two of the main strengths of this framework are the ability to deal with arbitrarily large or complicated parameter spaces, and to avoid any restriction on the theory calculations used, *e.g.* in relationship with the inclusion of higher-order EFT terms. As a proof of concept, SMEFiT was used in [23] to analyse about a hundred top quark production measurements from the LHC. In total, $N_{\mathrm{op}} = 34$ independent degrees of freedom at mass-dimension six were constrained simultaneously, including both linear, $\mathcal{O}\left(\Lambda^{-2}\right)$, and quadratic, $\mathcal{O}\left(\Lambda^{-4}\right)$, EFT effects as well as NLO QCD corrections.

As more experimental data becomes available, the probability distribution in the space of the SMEFT parameters should be correspondingly updated. This can naturally be achieved by performing a new fit to the extended set of data, which will however require in general a significant computational effort. In many situations, however, one would like to quantify the impact of a new measurement on the SMEFT parameter space more efficiently, *i.e.* without having to perform an actual fit. This may routinely happen whenever a new measurement is presented by the LHC experimental collaborations. In order to do so, one may wonder whether methods developed in order to quantify the PDF sensitivity to new data can help, such as the profiling of Hessian PDF sets [32, 33] or the Bayesian reweighting of Monte Carlo PDF sets [34, 35].

The aim of this work is to demonstrate that Bayesian reweighting, originally developed for Monte Carlo PDF sets in [34, 35], can be successfully extended to the SMEFiT framework. We do this as a proof of concept: given a prior SMEFiT fit based on a variant of our previous study [23], we show that single top-quark production measurements can be equivalently included in the prior either by Bayesian reweighting or by a new fit. We quantify the amount of new information that the measurements are bringing into the SMEFT parameter space by means of appropriate estimators, such as the Shannon entropy and the Kolmogorov-Smirnov statistic. We discuss the limitations of the method, explore the conditions under which it can be safely applied, and study its dependence upon a different definition of the replica weights, as proposed by Giele and Keller [36, 37] (see also [38]).

The outline of this paper is as follows. In Sect. 2 we review the Bayesian reweighting

method in the context of a SMEFiT analysis. In Sect. 3 we validate the method by reweighting a SMEFiT prior with different single-top datasets and by comparing the results with the corresponding fits. In Sect. 4 we study the sensitivity of the reweighting method upon alternative definitions of the weights. We summarise our findings in Sect. 5. Our results are made publicly available in the form of a stand-alone Python code, which we describe in the Appendix.

## 2 Bayesian reweighting revisited

Bayesian reweighting was originally developed in the case of PDFs in Refs [34,35], inspired by the earlier studies of [36,37]. It assumes that the probability density in the space of PDFs is represented by an ensemble of $N_{\text{rep}}$ equally probable Monte Carlo replicas

$$\{f_i^{(k)}(x, Q_0)\}, \quad i = 1, \ldots, n_f, \quad k = 1, \ldots, N_{\text{rep}}, \tag{2.1}$$

where $n_f$ is the number of active partons at the initial parametrisation scale $Q_0$ and $f$ is the corresponding PDF. The ensemble, Eq. (2.1), is obtained by sampling $N_{\text{rep}}$ replicas from the experimental data and then by performing a separate PDF fit to each of them.

Analogously, a SMEFiT analysis represents the probability density in the space of Wilson coefficients (or SMEFT parameters) as an ensemble of $N_{\text{rep}}$ Monte Carlo replicas

$$\{c_i^{(k)}\}, \quad i = 1, \ldots, N_{\text{op}}, \quad k = 1, \ldots, N_{\text{rep}}, \tag{2.2}$$

where $N_{\text{op}}$ is the number of independent dimension-6 operators $\{\mathcal{O}_i^{(6)}\}$ that define the fitting basis of the analysis and $c_i$ are the corresponding Wilson coefficients that enter the SMEFT Lagrangian,

$$\mathcal{L}_{\text{SMEFT}} = \mathcal{L}_{\text{SM}} + \sum_i^{N_{\text{op}}} \frac{c_i}{\Lambda^2} \mathcal{O}_i^{(6)}, \tag{2.3}$$

with $\Lambda$ the characteristic energy scale where new physics sets in. Since we neglect operator running effects [39], the coefficients $\{c_i^{(k)}\}$ are scale independent. The ensemble, Eq (2.2), can be obtained as a result of a fit in the SMEFiT framework.

The starting point of Bayesian reweighting is therefore a realisation of Eq. (2.2), which we will henceforth call the *prior*. The next step is to quantify the impact of some new measurement on the prior. Following Bayesian inference, this can be achieved by associating a weight $\omega_k$ to each Monte Carlo replica in the prior. The value of these weights depends on the agreement (or lack thereof) between the theory predictions constructed from each replica in the prior and the new dataset. Their analytic expression is [34,35]

$$\omega_k \propto \left(\chi_k^2\right)^{(n_{\text{dat}}-1)/2} \exp\left(-\chi_k^2/2\right), \quad k = 1, \ldots, N_{\text{rep}}, \tag{2.4}$$

where $n_{\text{dat}}$ is the number of data points in the new dataset and $\chi_k^2$ is the unnormalised $\chi^2$ of the new dataset computed with the $k$-th replica in the prior. These weights are normalised in such a way that their sum adds up to the total number of replicas, namely

$$\sum_{k=1}^{N_{\text{rep}}} \omega_k = N_{\text{rep}}. \tag{2.5}$$

We will discuss in Sect. 4 how results are affected if the Giele-Keller expression of the weights [36, 37], which differs from Eq. (2.4), is used instead. After the inclusion of the new data, replicas are no longer equally probable. The statistical features of the ensemble should therefore be

computed accordingly. For instance, the new expectation values are given by weighted means

$$\langle c_i \rangle = \frac{1}{N_{\text{rep}}} \sum_{k=1}^{N_{\text{rep}}} \omega_k \, c_i^{(k)}, \tag{2.6}$$

and likewise for other estimators such as variances and correlations.

In the definition of the weights, Eq. (2.4), the unnormalised $\chi_k^2$ associated to the $k$-th Monte Carlo replica is constructed as

$$\chi_k^2 = \sum_{i,j=1}^{n_{\text{dat}}} \left( \mathcal{F}_i^{(\text{th})}(\{c_k\}) - \mathcal{F}_i^{(\text{exp})} \right) (\text{cov}^{-1})_{ij} \left( \mathcal{F}_j^{(\text{th})}(\{c_k\}) - \mathcal{F}_j^{(\text{exp})} \right), \tag{2.7}$$

where $\mathcal{F}_i^{(\text{th})}(\{c_k\})$ is the theoretical prediction for the $i$-th cross section $\mathcal{F}_i$ evaluated using the Wilson coefficients associated to the k-th replica, $\{c_k\}$, and $\mathcal{F}_i^{(\text{exp})}$ is the central value of the corresponding experimental measurement. Note that in Eq. (2.7) the sum runs over only the data points of the new dataset that is being added by reweighting, while all the information from the prior fit is encoded in the Wilson coefficients $\{c_k\}$ associated to the corresponding Monte Carlo sample.

The total covariance matrix, $\text{cov}_{ij}$ in Eq. (2.7), should contain all the relevant sources of experimental and theoretical uncertainties. Assuming that theoretical uncertainties follow an underlying Gaussian distribution, and that they are uncorrelated to the experimental uncertainties, it can be shown [40] that

$$\text{cov}_{ij} = \text{cov}_{ij}^{(\text{exp})} + \text{cov}_{ij}^{(\text{th})}, \tag{2.8}$$

that is, the total covariance matrix is given by the sum of the experimental and theoretical covariance matrices. The experimental covariance matrix is constructed using the '$t_0$' prescription [41],

$$(\text{cov}_{t_0})_{ij}^{(\text{exp})} \equiv \left( \sigma_i^{(\text{stat})} \right)^2 \delta_{ij} + \left( \sum_{\alpha=1}^{N_{\text{sys}}} \sigma_{i,\alpha}^{(\text{sys})} \sigma_{j,\alpha}^{(\text{sys})} \mathcal{F}_i^{(\text{exp})} \mathcal{F}_j^{(\text{exp})} \right.$$
$$\left. + \sum_{\beta=1}^{N_{\text{norm}}} \sigma_{i,\beta}^{(\text{norm})} \sigma_{j,\beta}^{(\text{norm})} \mathcal{F}_i^{(\text{th},0)} \mathcal{F}_j^{(\text{th},0)} \right), \tag{2.9}$$

where 'sys' ('norm') indicates the additive (multiplicative) relative experimental systematic errors separately; $\mathcal{F}_i^{(\text{th},0)}$ corresponds to a fixed set of theoretical predictions obtained from a previous fit. All available sources of statistical and systematic uncertainties for a given dataset are considered in Eq. (2.9), including bin-by-bin correlations whenever available. The theoretical covariance matrix includes only the contribution from the PDF uncertainties in this analysis. It is given by

$$\text{cov}_{ij}^{(\text{th})} = \left\langle \mathcal{F}_i^{(\text{th})(\text{r})} \mathcal{F}_j^{(\text{th})(\text{r})} \right\rangle_{\text{rep}} - \left\langle \mathcal{F}_i^{(\text{th})(\text{r})} \right\rangle_{\text{rep}} \left\langle \mathcal{F}_j^{(\text{th})(\text{r})} \right\rangle_{\text{rep}}, \tag{2.10}$$

where the theoretical predictions $\mathcal{F}_i^{(\text{th})(\text{r})}$ are computed using the SM theory and the $r$-th replica from the NNPDF3.1NNLO no-top PDF set (which excludes all top quark measurements to avoid double counting).

After reweighting, replicas with small weights become almost irrelevant. This implies that the reweighted ensemble will be less efficient than the prior in representing the probability

distribution in the space of SMEFT parameters. This loss of efficiency is quantified by the Shannon entropy (or the effective number of replicas left after reweighting)

$$N_{\text{eff}} = \exp\left( \frac{1}{N_{\text{rep}}} \sum_{k=1}^{N_{\text{rep}}} \omega_k \ln \frac{N_{\text{rep}}}{\omega_k} \right). \tag{2.11}$$

This is the number of replicas needed in a hypothetical new fit to obtain an ensemble as accurate as the reweighted ensemble. If $N_{\text{eff}}$ becomes too low, the reweighting procedure no longer provides a reliable representation of the probability distribution in the space of SMEFT parameters. As a rule of thumb, in this work we require $N_{\text{eff}} \gtrsim 100$. This value was determined by studying the dependence of the SMEFiT results upon the number of Monte Carlo replicas in our previous study [23]. If $N_{\text{eff}} \lesssim 100$, either a prior set consisting of a larger number of starting replicas $N_{\text{rep}}$ or a new fit would be required to properly incorporate the information contained in the new data.

Such a situation can happen in two cases. First, if the new data contains a lot of new information, for example because it heavily constrains a new region of the parameter space or because it has a large statistical power. Second, if the new data is inconsistent with the old in the theoretical framework provided by the SMEFT. These two cases can be distinguished by examining the $\chi^2$ profile of the new data: if there are very few replicas with a $\chi^2$ per data point of order unity (or lower) in the reweighted ensemble, then the new data is inconsistent with the old. The inconsistency can be quantified by computing the $\mathcal{P}(\alpha)$ distribution (see [34] for further details), defined as

$$\mathcal{P}(\alpha) \propto \frac{1}{\alpha} \sum_{k=1}^{N_{\text{rep}}} \omega_k(\alpha), \tag{2.12}$$

where $\alpha$ is the factor by which the uncertainty on the new data must be rescaled to make them consistent with the old. If $\text{argmax}\,\mathcal{P}(\alpha) \sim 1$, the new data is consistent with the old; if $\text{argmax}\,\mathcal{P}(\alpha) \gg 1$, it is not.

A limitation of the effective number of replicas $N_{\text{eff}}$, Eq. (2.11), and of the $\mathcal{P}(\alpha)$ distribution, Eq. (2.12), is that they provide only a global measure of the impact of the new data. They do not allow one to determine which specific directions of the SMEFT operator space are being constrained the most. Such an information can be instead accessed by means of the Kolmogorov-Smirnov (KS) statistic. This estimator is defined as

$$\text{KS} = \sup_{\langle c_i \rangle} |\mathcal{F}_{\text{rw}}(\langle c_i \rangle) - \mathcal{F}_{\text{fit}}(\langle c_i \rangle)|, \tag{2.13}$$

*i.e.* as the supremum of the set of distances between the reweighted and the refitted probability distributions for each SMEFT operator, $\mathcal{F}_{\text{rw}}(\langle c_i \rangle)$ and $\mathcal{F}_{\text{fit}}(\langle c_i \rangle)$, respectively. Clearly $0 \leq \text{KS} \leq 1$: $\text{KS} \sim 0$ if the coefficients obtained either from reweighting or from a new fit belong to ensembles that represent the same probability distribution; $\text{KS} \to 1$ if they belong to ensembles that represent different probability distributions.

The transition between the two regimes is smooth. As an example, in Fig. 2.1 we show the value of the KS statistic, Eq. (2.13), between two Gaussian distributions sampled $N_{\text{rep}} = 10^4$ times each. One distribution (grey histogram) has mean $\mu_0 = 0$ and standard deviation $\sigma_0 = 1$, while the other distribution (green histogram) has mean $\mu_1 = \Delta\mu$ and standard deviation $\sigma_1 = \sigma_0 - \Delta\sigma$. The values of $\Delta\mu$ and of $\Delta\sigma$ are being increased from top to bottom and from left to right, respectively. While the KS statistic does not provide a clear-cut threshold to classify the two distributions as the same or not, it can be used as a guide to disentangle genuine effects of new data in the SMEFT operator space from statistical fluctuations. We should note that the examples shown in Fig. 2.1 are only valid for Gaussian distributions; in general, the probability distributions of Wilson coefficients can be non-Gaussian.

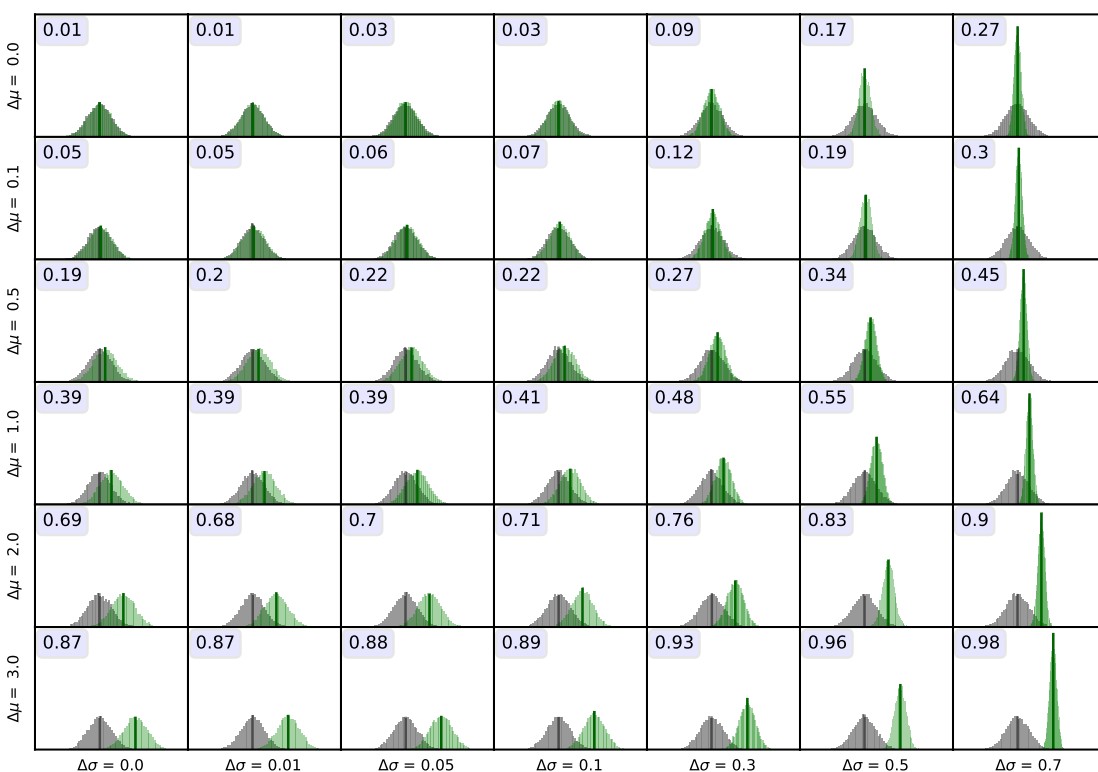

Figure 2.1: The value of the KS statistic, Eq. (2.13), between two Gaussian distributions sampled $N_{\text{rep}} = 10^4$ times each. The grey distribution has mean $\mu_0 = 0$ and standard deviation $\sigma_0 = 1$. The green distribution has mean $\mu_1 = \Delta\mu$ and standard deviation $\sigma_1 = \sigma_0 - \Delta\sigma$.

After reweighting, the prior is accompanied by a set of weights. For practical reasons, it is convenient to replace both of them with a new set of replicas which reproduce the reweighted probability distribution in the space of SMEFT parameters, but are again equally probable. This can be achieved by means of unweighting [35]. For statistical purposes, *e.g.* for the calculation of 95% confidence level (CL) intervals, the unweighted set can be treated in the same way as the prior (and as all sets obtained in a new fit).

## 3 Reweighting the SMEFT parameter space

We now explicitly illustrate how Bayesian reweighting works with a SMEFT Monte Carlo fit. We first describe our choice of prior for Eq. (2.2) and we reweight and unweight it with several sets of single-top production data. We then monitor the efficiency loss of the reweighted set and we verify under which conditions reweighting leads to results equivalent to those of a new fit. We finally test such conditions upon variation of the process type used to reweight the prior.

Table 3.1: The measurements of single-top quark production at the LHC (both in the $t$-channel and in the $s$-channel) used in this analysis to validate the results of Bayesian reweighting. For each dataset, we indicate the dataset label, the center of mass energy $\sqrt{s}$, the production mechanism, the type of observables, the number of data points $n_{\text{dat}}$, and the publication reference.

| ID | Dataset | $\sqrt{s}$ | Info | Observables | $n_{\text{dat}}$ | Ref. |
|----|---------|------------|------|-------------|------------------|------|
| 1 | CMS_t_tch_8TeV_dif | **8 TeV** | $t$-channel | $d\sigma/d\lvert y^{(t+\bar{t})}\rvert$ | 6 | [42] |
| 2 | ATLAS_t_tch_8TeV | **8 TeV** | $t$-channel | $d\sigma(t)/dy_t$ | 4 | [43] |
| 3 | ATLAS_t_tch_8TeV | **8 TeV** | $t$-channel | $d\sigma(\bar{t})/dy_{\bar{t}}$ | 4 | [43] |
| 4 | CMS_t_tch_13TeV_dif | **13 TeV** | $t$-channel | $d\sigma/d\lvert y^{(t+\bar{t})}\rvert$ | 4 | [44] |
| 5 | CMS_t_tch_8TeV_inc | **8 TeV** | $t$-channel | $\sigma_{\text{tot}}(t), \sigma_{\text{tot}}(\bar{t})$ | 2 | [48] |
| 6 | CMS_t_tch_13TeV_inc | **13 TeV** | $t$-channel | $\sigma_{\text{tot}}(t+\bar{t})$ | 1 | [49] |
| 7 | ATLAS_t_tch_8TeV | **8 TeV** | $t$-channel | $\sigma_{\text{tot}}(t), \sigma_{\text{tot}}(\bar{t})$ | 2 | [43] |
| 8 | ATLAS_t_tch_13TeV | **13 TeV** | $t$-channel | $\sigma_{\text{tot}}(t), \sigma_{\text{tot}}(\bar{t})$ | 2 | [45] |
| 9 | ATLAS_t_sch_8TeV | **8 TeV** | $s$-channel | $\sigma_{\text{tot}}(t+\bar{t})$ | 1 | [47] |
| 10 | CMS_t_sch_8TeV | **8 TeV** | $s$-channel | $\sigma_{\text{tot}}(t+\bar{t})$ | 1 | [46] |

## 3.1 Choice of prior and of reweighting datasets

We choose the prior for Eq. (2.2) as a fit obtained in the SMEFiT framework from our previous work [23]. Specifically we consider a variant of our baseline result, where measurements of inclusive single-top quark production in the $t$-channel [42–45] and in the $s$-channel [46, 47] are removed from the default dataset: in total $n_{\text{dat}} = 20$ data points for single-top $t$-channel production (total and differential cross sections) and $n_{\text{dat}} = 2$ data points (total cross sections) for single-top $s$-channel production. The prior is thus based on $n_{\text{dat}} = 81$ data points (the 103 used in the baseline fit of [23] minus the above 22).

To ensure a sufficiently accurate representation of the probability distribution in the SMEFT parameter space, the prior is made of $N_{\text{rep}} = 10^4$ Monte Carlo replicas. Such a large sample – one order of magnitude larger than the sample used in [23] – is required to mitigate the efficiency loss upon reweighting. The effective number of replicas, Eq. (2.11), would otherwise become too small and reweighted results will no longer be reliable.

The prior is then reweighted with the datasets of single-top production listed in Table 3.1. These sets include all the sets originally removed from the default fit in [23] to generate the prior. Each of them is labelled as in our previous work (see Table 3.3. in [23]). In addition, we consider three extra datasets for the total cross sections from CMS at 8 and 13 TeV and from ATLAS at 8 TeV (for a total of $n_{\text{dat}} = 5$ data points). This data was not taken into account in the fit of Ref. [23] to avoid a double-counting issue, since we already included the corresponding absolute differential distributions determined from the same data taking. We believe that this is not an issue here, since our aim is to validate the reweighting procedure rather than to extract accurate bounds on the SMEFT parameters. We therefore retain all the datasets collected in Table 3.1.

We reweight the prior with the data in Table 3.1 either sequentially (by adding one dataset after the other) or simultaneously (by adding all the datasets at once). In the first case, we

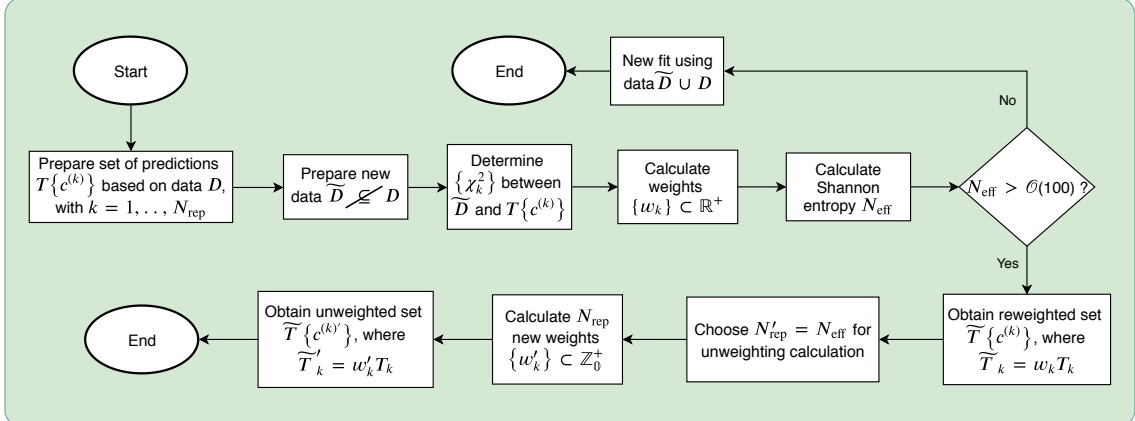

Figure 3.1: Overview of the reweighting/unweighting procedure. The procedure is successful if the probability distribution associated to the unweighted set coincides with the one from a new fit.

monitor the efficiency loss and quantify the constraining power of each dataset. In the second case, we validate the goodness of reweighting against the results of a new fit to the extended datasets, by checking that they lead to equivalent results (within statistical fluctuations). Our strategy is schematically summarised in Fig. 3.1.

## 3.2 Monitoring the efficiency loss: the effective number of replicas

We first reweight our prior by including sequentially one dataset after the other, following the order given in Table 3.1. In Fig. 3.2 we show the value of the effective number of replicas $N_{\mathrm{eff}}$, Eq. (2.11), for each step: point "0" corresponds to the prior, which does not contain any of the single-top production measurements listed in Table 3.1; points "1"-"8" correspond to the sets reweighted with each of the single-top t-channel datasets; and point "9" corresponds to the set further reweighted with the total single-top s-channel production cross section from ATLAS at 8 TeV. Reweighting with the remaining total single-top s-channel production cross section from CMS at 8 TeV would in principle correspond to an extra point on the right of Fig. 3.2. However it is not displayed because the efficiency loss of the reweighted ensemble is already significant ($N_{\mathrm{eff}} \lesssim 100$) for point "9". Therefore the corresponding results cannot be trusted.

From Fig. 3.2 one observes that the original number of replicas in the prior, $N_{\mathrm{rep}} = 10^4$, are reduced to $N_{\mathrm{eff}} \simeq 550$ effective replicas once the first single-top t-channel production dataset is added. The subsequent addition of the remaining t-channel measurements leads to a further, but mild, decrease of the value of $N_{\mathrm{eff}}$ down to around $N_{\mathrm{eff}} \simeq 300$. This behaviour can be understood if we consider that, the first time one adds a single-top t-channel dataset, one is constraining several directions in the parameter space that had large uncertainties or were degenerate in the prior. Adding subsequent measurements of the same type only refines the constraints provided by this first dataset.

One may wonder whether the initial abrupt decrease in the effective number of replicas is just a consequence of the fact that the specific dataset is inconsistent with the prior. Computing the $\mathcal{P}(\alpha)$ distribution rules out this possibility as expected: we know from our previous work [23] that all the datasets in Table 3.1 are consistent with the prior. For this reason we will refrain from showing the $\mathcal{P}(\alpha)$ distribution in the sequel.

Our understanding is further confirmed by observing that once the s-channel measurements are subsequently added, then $N_{\mathrm{eff}}$ falls from $\simeq 300$ to below 50. Again, there is a large amount of information being added into the probability distribution once a completely new type of process is added, since now one becomes sensitive to new combinations of SMEFT

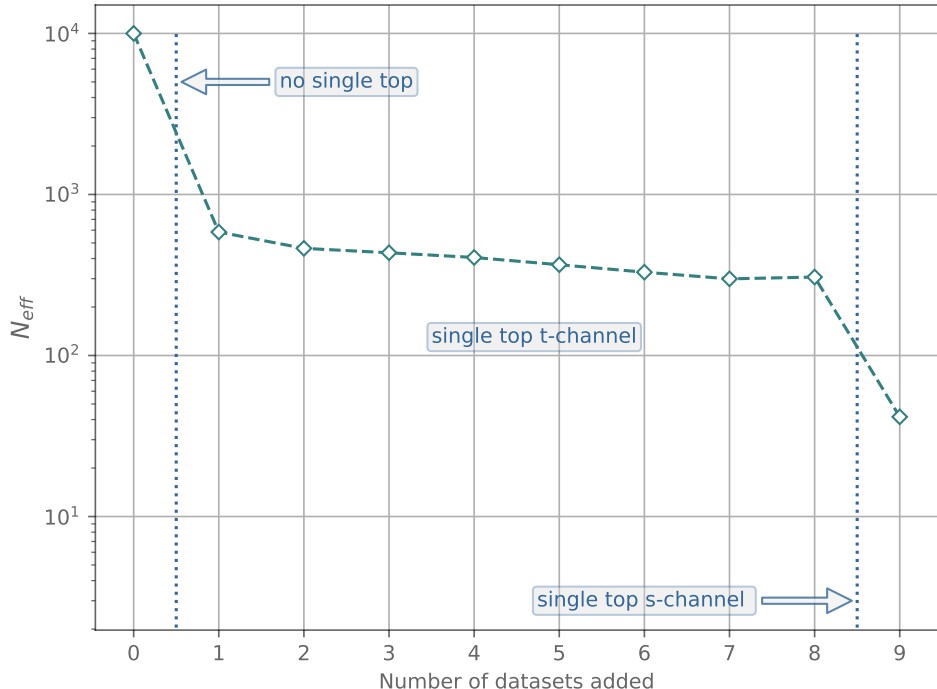

Figure 3.2: The value of the effective number of replicas $N_{\text{eff}}$, Eq. (2.11), in the prior, which does not contain any of the single-top production measurements listed in Table 3.1, and once the various single-top datasets are sequentially added by reweighting. As indicated in the plot, first we add the $t$-channel datasets and then the $s$-channel datasets following the order given in Table 3.1.

parameters that are unconstrained by the measurements previously considered. Given that $N_{\text{eff}} \simeq 50$, the reliability of the reweighting method in this case would be questionable. Including both the $t$- and the $s$-channel measurements by reweighting would require a prior based on a much larger number of replicas, *e.g.* $N_{\text{rep}} = \mathcal{O}(10^5)$.

Finally we have also verified that the order in which specific datasets are being added via reweighting does not modify the pattern observed in Fig. 3.2 nor the final result of the procedure. This behaviour is consistent with what was found in the PDF case [34, 35].

### 3.3 Validation of reweighting: single-top $t$-channel data

We now reweight our prior by including simultaneously all the single-top $t$-channel datasets at once. For the time being, we do not consider single-top $s$-channel datasets, because this will result in a too large efficiency loss (see above). Our aim is to validate the reweighting procedure by comparing the resulting probability distribution with that obtained from a new fit to the same datasets.

In the upper panel of Fig. 3.3 we compare the results obtained from reweighting and from the new fit. Specifically we show the 95% CL bounds for the $N_{\text{op}} = 34$ Wilson coefficients considered in our previous SMEFT analysis of the top quark sector [23]. Note that we are assuming that $\Lambda = 1$ TeV. For completeness, we also show the corresponding unweighted results: in all cases we find excellent agreement with the reweighted results. We will thus treat them as equivalent in the following.

From this comparison one finds that the results obtained from reweighting or from a new fit are reasonably similar in most cases. To facilitate their interpretation, we compare the relative

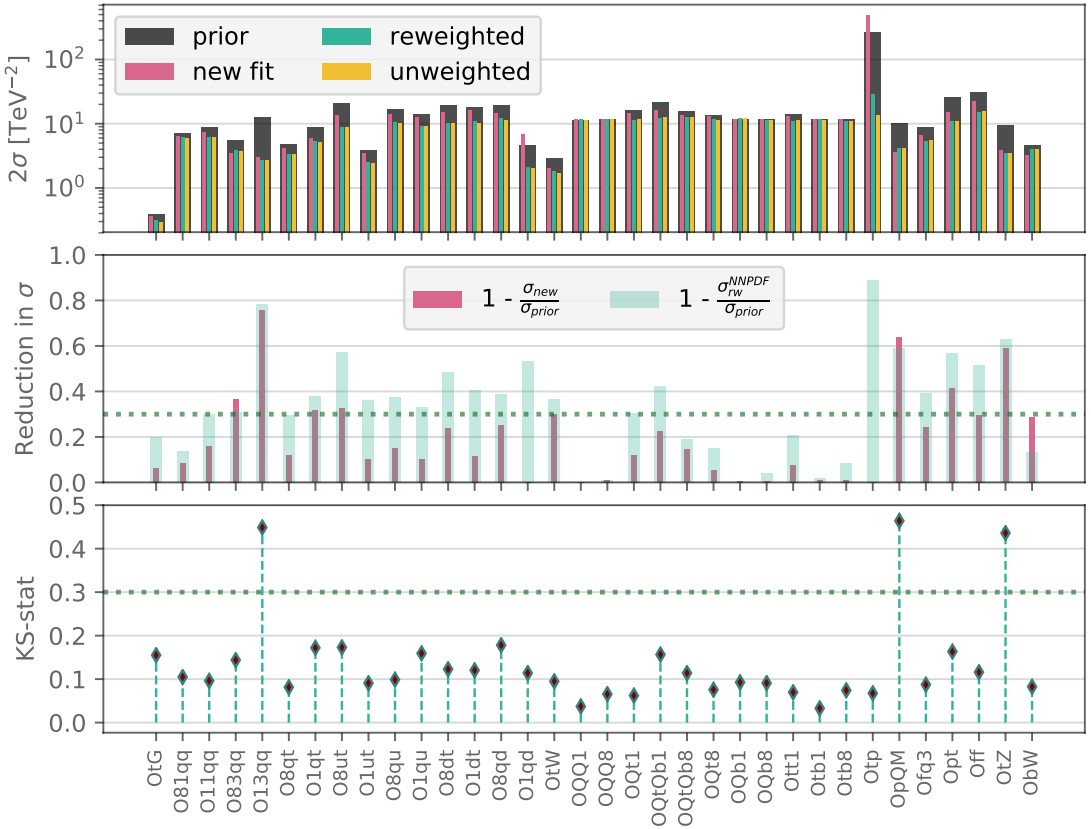

Figure 3.3: Upper panel: the 95% CL bounds for the $N_{\text{op}} = 34$ Wilson coefficients considered in this SMEFT analysis of the top quark sector. We compare the prior results (without any $t$- or $s$-channel single-top production data included) with those after the $t$-channel measurements have been added either by reweighting or by performing a new fit. Central panel: the relative 68% CL uncertainty reduction compared to the prior, both for the reweighted and for the new fit cases. Lower panel: the associated value of the KS statistic computed between the unweighted and the prior results. In both the central and lower panels, the horizontal dotted lines indicate the thresholds to select the operators for which Bayesian reweighting is meaningful.

68% CL uncertainty reduction between the reweighted and the prior cases, $1 - \sigma_{\text{rw}}^{\text{NNPDF}}/\sigma_{\text{prior}}$, and between the new fit and the prior cases, $1 - \sigma_{\text{new}}/\sigma_{\text{prior}}$. We observe that for three degrees of freedom the reweighting of the prior with the $t$-channel single-top cross section data leads to a reduction of the uncertainties larger than a factor of two, consistently with the new fit. These are the Wilson coefficients associated to the O13qq, OpQM and OtZ SMEFT operators (we will henceforth use the notation of [23]). Not surprisingly these are the three operators for which adding $t$-channel single-top data to the prior has the largest effect. In particular, O13qq and OpQM are either directly (or indirectly, via correlations with other coefficients) constrained by $t$-channel single-top data. The reason OtZ is also more constrained is because the data either provides access to a previously unconstrained direction in the SMEFT parameter space, see Table 3.5 in [23], or because it breaks degeneracies between directions. We therefore conclude that reweighting leads to results equivalent to those of a new fit for the operators that are being more directly constrained by the new data.

If we now look at other operators, we still clearly find that reweighting leads to a reduction of the 95% CL bounds in comparison to the prior. However such a reduction seems sometimes over-optimistic, especially if it is compared to the new fit results (see for example the Otp or

O1qd operators). In this case, reweighting seemingly fails. Nevertheless the 95% CL bounds are only a rough measure of the actual change in the probability distributions from the prior to the reweighted ensemble. Their interpretation is particularly unclear when statistical fluctuations (including from finite-size effects intrinsic to a Monte Carlo analysis such as the current one) become large. This mostly happens for poorly-constrained operators.

To discriminate whether the discrepancies observed in Fig. 3.3 are induced by a genuine change in the probability distributions of the various operators or by a statistical fluctuation, we look at the corresponding KS statistic, Eq. (2.13). Only when the value of the KS statistic is sufficiently large, can one claim that the differences between the prior and the reweighted distributions are statistically significant. With this motivation in mind, in the lower panel of Fig. 3.3 we display, for each Wilson coefficient, the values of the KS statistic computed between the unweighted and the prior distributions. The largest values of the KS statistic are associated to the O13qq, OpQM and OtZ operators. These are precisely the operators for which we know that the data has the largest effect and for which reweighting is equivalent to a new fit. Low values of the KS statistic are associated to most of the other operators, including those for which the reduction of the 95% CL bounds induced by reweighting is seemingly large (and even much larger than the reduction induced by a new fit). This is the case, *e.g.*, for the O1qd and Otp operators.

The results of Fig. 3.3 show that, in a global SMEFT analysis, reweighting successfully reproduces the results of a new fit when the two following conditions are satisfied:

- the size of 95% CL intervals of a specific operator after reweighting is reduced by an amount higher than a given threshold;

- the value of the KS statistic is sufficiently high to ensure that the modification in the probability distributions is not induced by a statistical fluctuation.

Of course there will always be some ambiguity when setting the thresholds for the 95% CL bound reductions and for the KS statistic. In Fig. 3.3 we indicate two possible values of these thresholds in the central and lower panels with dotted horizontal lines: $1 - \sigma_{\mathrm{rw}}^{\mathrm{NNPDF}}/\sigma_{\mathrm{prior}} = 1 - \sigma_{\mathrm{new}}/\sigma_{\mathrm{prior}} = 0.3$ and KS = 0.3. These values will select O13qq, OpQM and OtZ as the only three out of the $N_{\mathrm{op}} = 34$ operators for which the reweighted results are reliable. For the remaining operators, the two conditions above will not be satisfied and the corresponding reweighting results could not be trusted. While the selection criteria adopted here are mostly based on phenomenological evidence, it would be advantageous to derive more formal criteria that could be used in general situations. We defer such an investigation to future work.

As a final cross-check, the probability distributions of the three operators associated to the Wilson coefficients for which reweighting is applicable, O13qq, OpQM and OtZ, are displayed in Fig. 3.4. The prior results are compared to the results obtained by reweighting and unweighting the prior with the $t$-channel single-top production cross section data and the results obtained from a new fit to the same set of data. The prior and the new fit sets are made of $N_{\mathrm{rep}} = 10^4$ replicas; the unweighted set is made from $N_{\mathrm{eff}} = 300$ effective replicas. First, we observe how the prior distribution is significantly narrowed once the new data is added, either by reweighting or by a new fit, consistently with the results displayed in the central panel of Fig. 3.3. Second, we observe good agreement between the reweighted and the new fit shapes of the probability distributions, despite the former being based on a much smaller number of replicas than the latter. All this is consistent with the high value of the KS statistic associated to the three operators under examination.

In this work we have so far validated the reweighting approach using theory calculations that included $\mathcal{O}(\Lambda^{-4})$ corrections in the prior fit, in the reweighted results and in the new fit.

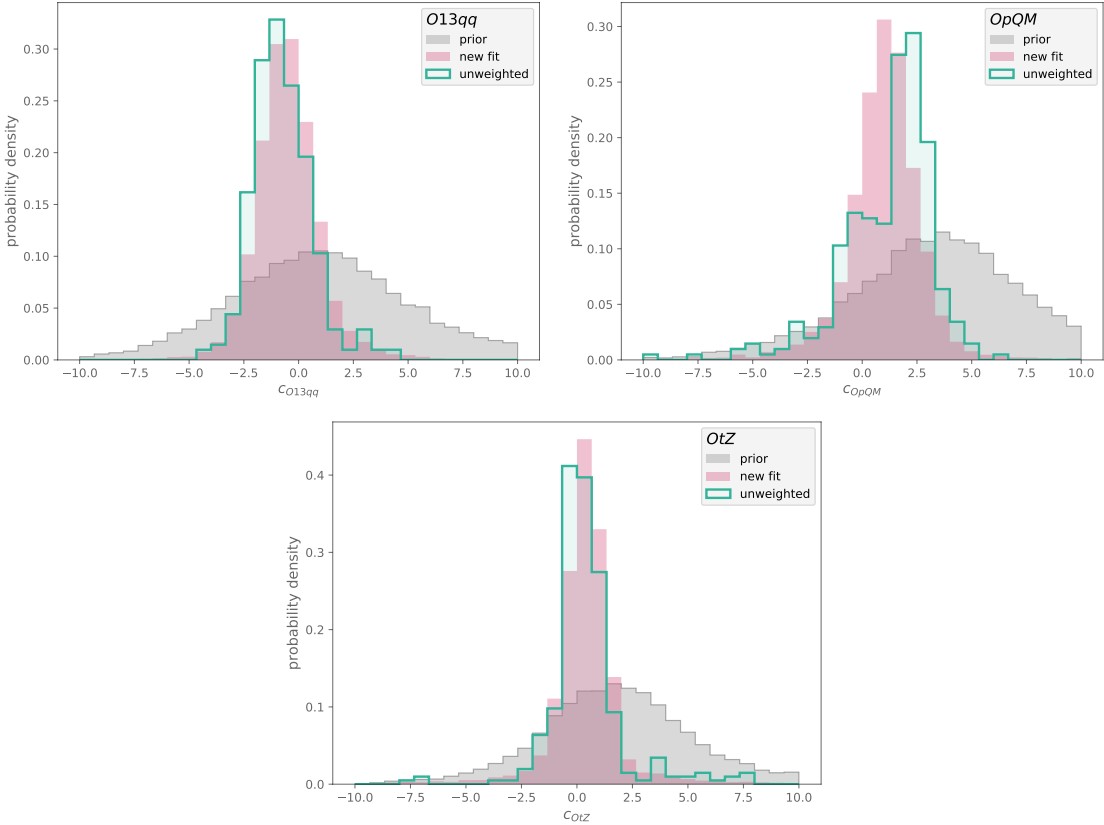

Figure 3.4: The probability distribution associated to the Wilson coefficients $c_{qq}^{13}, c_{pQM}$, and $c_{tZ}$. The prior results are compared to the results obtained by reweighting and unweighting the prior with the $t$-channel single-top production cross sections and from a new fit to the extended datasets.

However, the validity of Bayesian reweighting should be independent of the specific theory assumptions used. In this respect, we have verified that an agreement between reweighting and the new fit similar to that reported in Fig. 3.3 is obtained when only $\mathcal{O}\left(\Lambda^{-2}\right)$ corrections are included in the SMEFT theory calculation. This exercise demonstrates that Bayesian reweighting can be applied in the same way in both cases. It is up to the users to decide on their preferred theory settings — our SMEFiT results are available both at $\mathcal{O}\left(\Lambda^{-2}\right)$ and at $\mathcal{O}\left(\Lambda^{-4}\right)$.

Note that, in general, the subset of operators which are more affected by the new data differ between the $\mathcal{O}\left(\Lambda^{-2}\right)$ and the $\mathcal{O}\left(\Lambda^{-4}\right)$ fits. In the case of the $t$-channel single top production measurements, the three operators for which the impact of the new data is more marked are: O13qq, OpQM, and OtZ at $\mathcal{O}\left(\Lambda^{-4}\right)$; O13qq, OpQM, OtW and Opt at $\mathcal{O}\left(\Lambda^{-2}\right)$. The fact that the specific operators which are more constrained by a new piece of experimental information depend on whether the calculation is performed at $\mathcal{O}\left(\Lambda^{-2}\right)$ or the $\mathcal{O}\left(\Lambda^{-4}\right)$ is expected since in general each calculation probes different regions of the parameter space, as discussed in [23].

A final remark concerns the interpretation of the results presented in Fig. 3.3. Within a SMEFT analysis one is only sensitive to the ratio $c_k/\Lambda^2$ rather than to the absolute New Physics scale $\Lambda$. While here we assume $\Lambda = 1$ TeV for illustrative purposes, it is possible to interpret our results for any other value of $\Lambda$. In particular, the upper (lower) bounds on the $k$-th Wilson coefficient, $\delta c_k^+ \, (\delta c_k^-)$, should be rescaled as

$$\delta \widetilde{c}_k^{\pm} = \delta c_k^{\pm} \times \left(\frac{\widetilde{\Lambda}}{\Lambda}\right)^2 \tag{3.1}$$

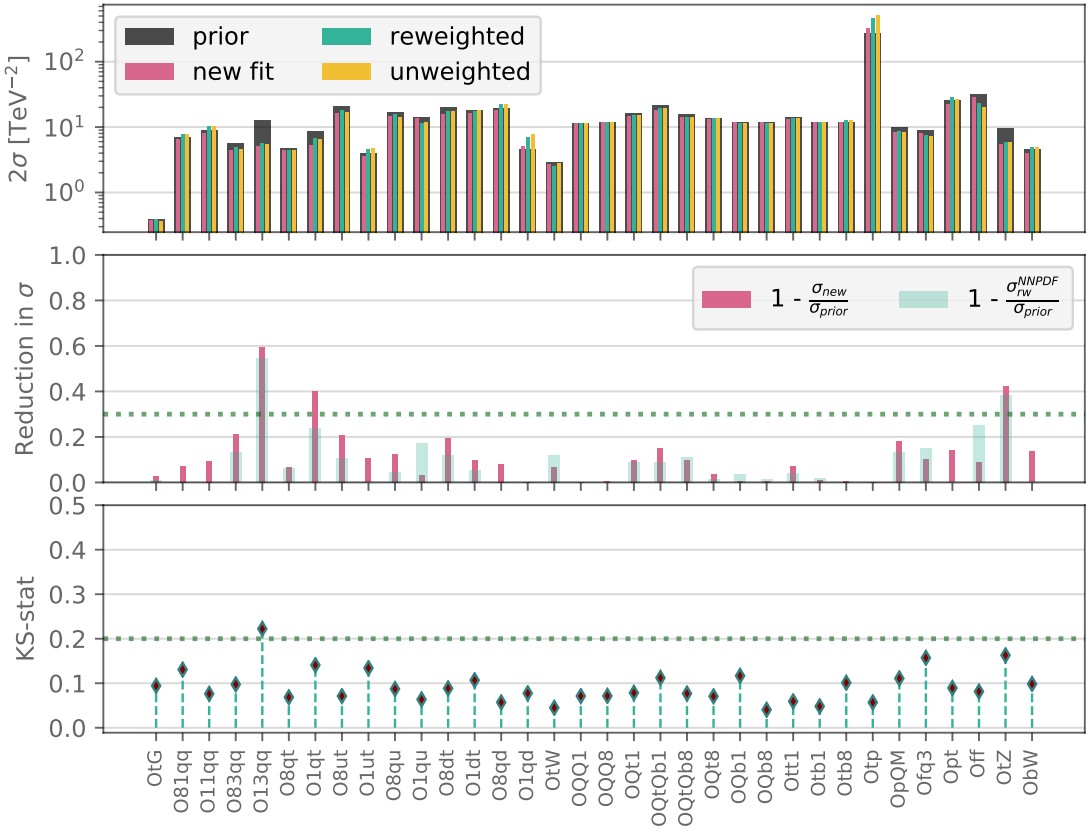

Figure 3.5: Same as Fig. 3.3, but in the case of single-top *s*-channel production total cross sections at 8 TeV. The selection threshold for the KS statistic is set to 0.2 in this case.

in comparison to the results shown here for the case $\widetilde{\Lambda} \neq \Lambda = 1$ TeV. This said, the validity of Bayesian reweighting is independent from the interpretation of the results in terms of a specific value of $\Lambda$. The important discussion about which values of $\Lambda$ can be probed when interpreting the results to ensure the validity of the EFT regime, see for instance [23] and references therein, should thus be separated from the validation of Bayesian reweighting.

## 3.4   Independence from the process type: single-top *s*-channel data

We now repeat the exercise carried out in the previous subsection by simultaneously reweighting our prior with all the single-top *s*-channel data listed in Table 3.1, *i.e.* the two cross sections from ATLAS and CMS at 8 TeV. Despite having only $n_{dat} = 2$ additional data points, one can in principle expect to improve the prior by a significant amount because the new process probes different top quark couplings with respect to those already included in the prior. By doing so, our purpose is to check whether the conclusions reached in the case of single-top *t*-channel datasets can be extended to datasets for processes of a different type.

In Fig. 3.5 we display the same results as in Fig. 3.3, but for the case of *s*-channel single-top production total cross sections. The impact of the $n_{dat} = 2$ *s*-channel data points is rather smaller than the impact of the $n_{dat} = 25$ *t*-channel ones, though still appreciable. The values of the KS statistic are consequently small for all the $N_{op} = 34$ operators but one: O13qq. In this case, this operator is the only one that satisfies the selection criteria defined above, whereby KS $\geq 0.2$ and $1 - \sigma_{rw}^{NNPDF}/\sigma_{prior} = 1 - \sigma_{new}/\sigma_{prior} \geq 0.3$. As expected, good agreement is found between the reweighted and the new fit results for the O13qq operator.

As discussed above there is an irreducible ambiguity in the choice of the threshold values for

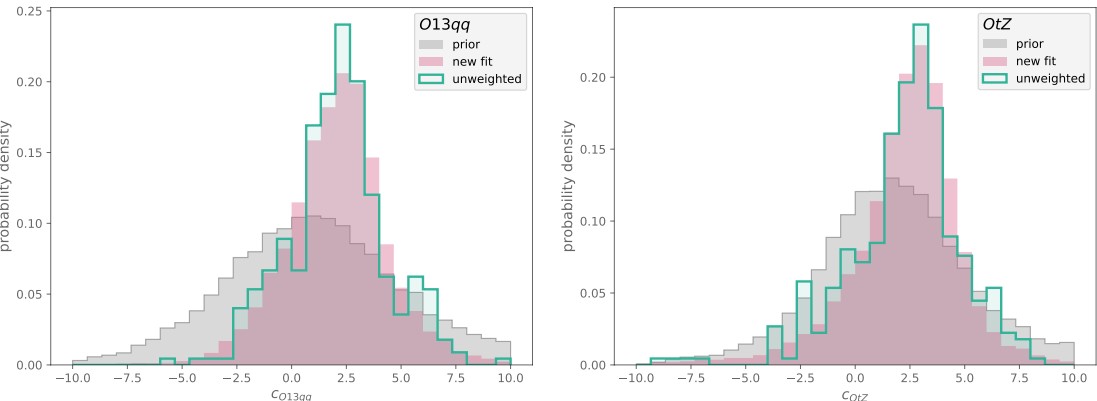

Figure 3.6: Same as Fig. 3.4 but for the Wilson coefficients mostly constrained by single-top $s$-channel production cross sections.

the relative reduction of the 95% CL bounds and of the KS statistic. For instance, if we look at the `OtZ` operator, the inclusion of the single-top $s$-channel cross sections leads to an uncertainty reduction of around 40% and to a KS statistic of KS $\simeq 0.18$; reasonable agreement between reweighted and new fit results is also observed for `OtZ`. However, this operator does not pass validation if we require KS $\geq 0.3$. Therefore it is up to the user to decide how conservative he wants to be: the higher the selection thresholds, the more reliable the reweighting results will be.

Finally in Fig. 3.6 we repeat the comparison shown in Fig. 3.4, but for the probability distributions of the Wilson coefficients associated to the `O13qq` and `OtZ` SMEFT operators. We show the results obtained from the prior, from reweighting and unweighting it with the single-top $s$-channel production cross sections and from a new fit to the same dataset. We find good agreement between the unweighted and the new fit results and observe how the relative narrowing of the distribution is less marked than in the case of single-top $t$-channel cross sections. This is understood as single-top $s$-channel measurements have less constraining power than single-top $t$-channel measurements once included in the prior.

## 4 Dependence on the choice of weights

The results presented in the previous section are based on the expression for the weights given by Eq. (2.4). This formula was originally derived in [34, 35] and has been routinely used to quantify the impact of new data in studies of PDFs since then. Results obtained with Eq. (2.4) were found to be equivalent to the results obtained with a new fit of the data in all cases, and were even benchmarked in a closure test [27]. In this work, we showed that Eq. (2.4) works also in the case of a global SMEFT analysis, provided specific selection criteria for the individual operators are satisfied. In this section, we study the dependence of our results upon the choice of weights.

### 4.1 Giele-Keller weights

An expression for the weights different from the one in Eq. (2.4) has been suggested in the past. Specifically, in a formulation which pre-dates the one in [34, 35], Giele and Keller advocated [36, 37] that the weights should read instead

$$\omega_k \propto \exp\left(-\chi_k^2/2\right), \quad k = 1, \ldots, N_{\text{rep}}, \tag{4.1}$$

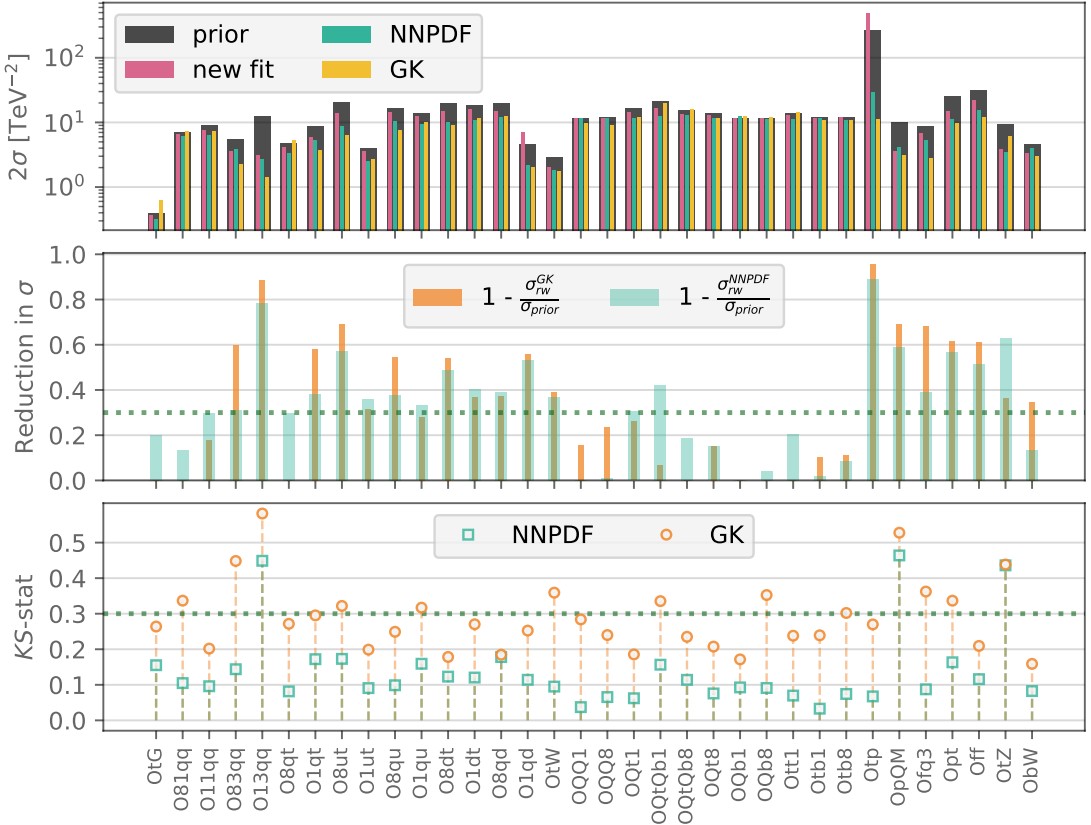

Figure 4.1: Same as Fig. 3.3, but for reweighted results obtained either with the NNPDF or the GK weights.

where, in comparison to Eq. (2.4), the prefactor $(\chi_k^2)^{(n_{\text{dat}}-1)/2}$ is dropped. We will refer to Eq. (2.4) and to Eq. (4.1) as NNPDF and GK weights, respectively, in the remainder of this section. The main difference between NNPDF and GK weights is that, for consistent data, the largest weights are assigned respectively to replicas either with $\chi_k^2 \simeq n_{\text{dat}}$ or with $\chi_k^2 \to 0$. If Eq. (4.1) is used instead of Eq. (2.4), a replica associated to $\chi_k^2 \to 0$ is not treated as an outlier and discarded, as it should, but it is assigned a large weight.

In order to explore the dependence of the reweighted results on the choice of the weight formula, we repeat the exercise presented in the previous section by using the GK weights instead of the NNPDF weights. By comparison with our previous results, we expect to determine whether the GK formula reproduces the results of a new fit as well as the NNPDF formula, and if it does so more efficiently. In principle, for $N_{\text{rep}} \to \infty$, it is conceivable that the NNPDF and the GK formulæ lead to indistinguishable results, which then become different only because of finite-size effects.

In Fig. 4.1 we compare the results obtained by reweighting the prior with all the single-top $t$-channel data in Table 3.1 either with the NNPDF or the GK weight formula. The format of the results is the same as in Fig. 3.3, *i.e.* each panel displays, from top to bottom, the 95% CL bounds, the corresponding relative reduction with respect to the prior and the KS statistic.

We recall that the reweighted results obtained with the NNPDF weights reproduce the results obtained with a new fit only for the three operators that are more directly constrained by the new data: O13qq, OpQM and OtZ. A meaningful comparison with the results obtained with GK weights should therefore first focus on these three operators. By inspection of Fig. 4.1 such a comparison reveals that NNPDF and GK results can be rather different. In particular,

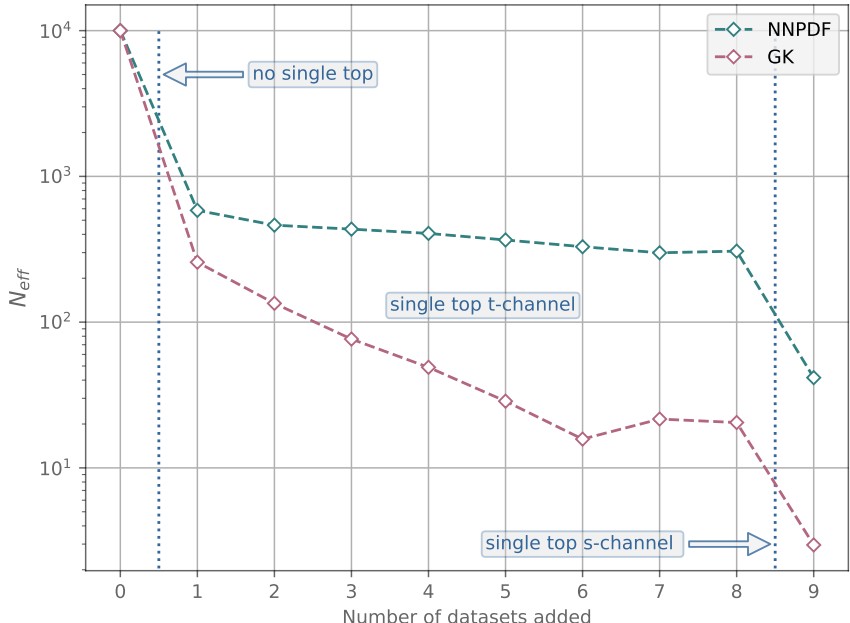

Figure 4.2: Same as Fig. 3.2, now comparing the effective number of replicas $N_{\text{eff}}$ upon the addition of new data using either the NNPDF or the GK weights. In both cases, the prior is the same and it is made of $N_{\text{rep}} = 10^4$ replicas.

the GK results can either overestimate (for `O13qq` and `OpQM`) or underestimate (for `OtZ`) the uncertainty reduction.

Marked differences between NNPDF and GK results persist across all the operators. The values of the KS statistic is consistently larger in the GK case than in the NNPDF case. We therefore investigate the behaviour of the efficiency in the GK case. In Fig. 4.2 we show the dependence of the effective number of replicas $N_{\text{eff}}$ upon the addition of new data both in the NNPDF and in the GK cases. Given that the same amount of new information is added, it is apparent that the effective number of replicas decreases much faster for GK weights than for NNPDF weights. This downwards trend persists even when more datasets of the same type are added to the prior. Such a behaviour is not as marked in the NNPDF case, where instead the reduction of the effective number of replicas after the inclusion of the first dataset of a given type is only mild.[1]

Moreover, from Fig. 4.2 one finds that, after the prior is reweighted with all the $t$-channel single-top cross sections in Table 3.1, $N_{\text{eff}} \simeq 300$ and $N_{\text{eff}} \simeq 20$, respectively, in the NNPDF and GK cases. In the latter case the effective number of replicas is simply too low for us to trust the reweighted results. The results of Fig. 4.1 should therefore be interpreted with care. Discrepancies between the NNPDF and GK cases could be explained as genuine differences between the corresponding weights, or else as large statistical fluctuations due to finite-size effects. This ambiguity is further illustrated in Fig. 4.3, where we compare the probability distribution of the `O13qq` and `OtZ` operators in the prior and in the reweighted sets obtained both in the NNPDF and in the GK cases. Differences between the reweighted NNPDF and GK distributions are apparent, as is the fact that the GK distribution lacks sufficient statistics to be reliable.

---

[1]In Fig. 4.2 the value of $N_{\text{eff}}$ increases slightly between datasets 6 and 7 for the GK case. When adding new measurements, the value of $N_{\text{eff}}$ should always decrease (or remain constant) up to statistical fluctuations. These fluctuations are negligible when the number of starting replicas is large enough, but not when one has only $N_{\text{eff}} \simeq 20$ replicas as in this specific case.

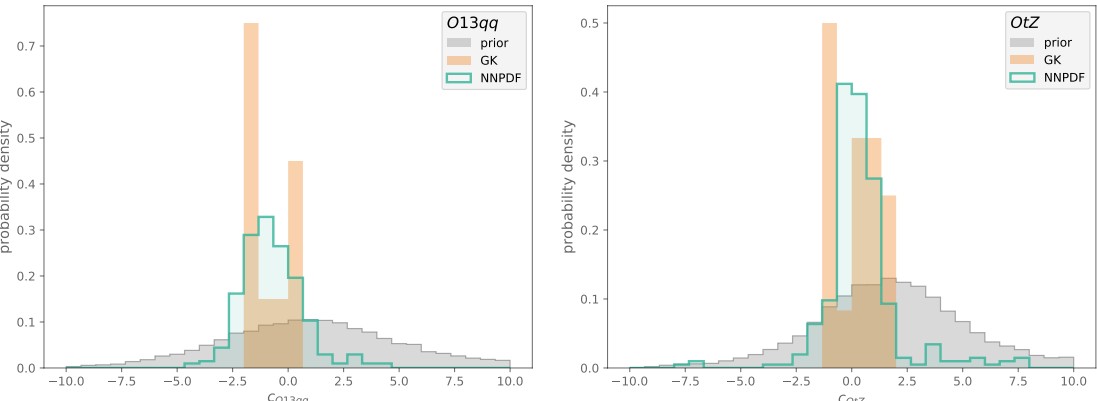

Figure 4.3: Same as Fig. 3.4 for the distributions of the `O13qq` and `OtZ` operators obtained from the prior and the reweighted sets with either the NNPDF or the GK weights. The number of effective replicas in the two cases is $N_{\mathrm{eff}} \simeq 300$ and $N_{\mathrm{eff}} \simeq 20$, respectively.

In order to understand whether the differences between the results obtained with NNPDF and GK weights can be explained as finite-size effects, we need to compare results with a sufficiently large effective number of replicas. A quick inspection of Fig. 4.2 reveals that this can be achieved for both NNPDF and GK when only the first dataset is added. In such a case, one would end up with $N_{\mathrm{eff}} \simeq 600$ and $N_{\mathrm{eff}} \simeq 250$ effective replicas, respectively.

In Fig. 4.4 we repeat the comparison shown in Fig. 4.1, now obtained with the inclusion of the first single-top $t$-channel data point only. Good agreement between the NNPDF and the GK reweighting results are now found for the three usual operators `O13qq`, `OpQM`, and `OtZ`. For other operators, residual differences are significantly reduced in comparison to Fig. 4.1; most notably, the NNPDF and GK values of the KS statistic are now much more consistent between each other.

Finally, in Fig. 4.5 we repeat the comparison between the prior, NNPDF and GK probability distributions of the Wilson coefficients associated to the `O13qq` and `OtZ` operators when only the first single-top $t$-channel data point is included. Now that finite-size effects are under control, good agreement is found between the NNPDF and GK reweighted distributions. Such an agreement is consistent with Fig. 4.4.

We find a very similar pattern of results if we repeat the above exercises with the single-top $s$-channel datasets. We can therefore conclude that, based on the phenomenological exploration presented in this study, for those SMEFT operators that satisfy our selection requirements, and provided that the efficiency loss is not too severe (that is, the effective number of replicas is large enough), using either the NNPDF or the GK weights leads to equivalent results. Under the above conditions, these results agree with a corresponding new fit. In general, however, GK weights appear to be rather less efficient than NNPDF weights. This behaviour could easily lead to misleading results, unless one is careful in monitoring their dependence on the effective number of replicas. In particular using the GK weights without ensuring that $N_{\mathrm{eff}}$ is sufficiently large might result in an overestimate of the impact that new measurements have on the SMEFT parameter space.

## 4.2 Hybrid weights

The rationale for comparing the results of the NNPDF and GK weights arises because in both cases there are studies that claim that one of the two is the correct expression based on formal arguments. In principle it might be possible that there exists an intermediate expression for the weights $\omega_k$ that interpolates between the NNPDF and GK weights while exhibiting a

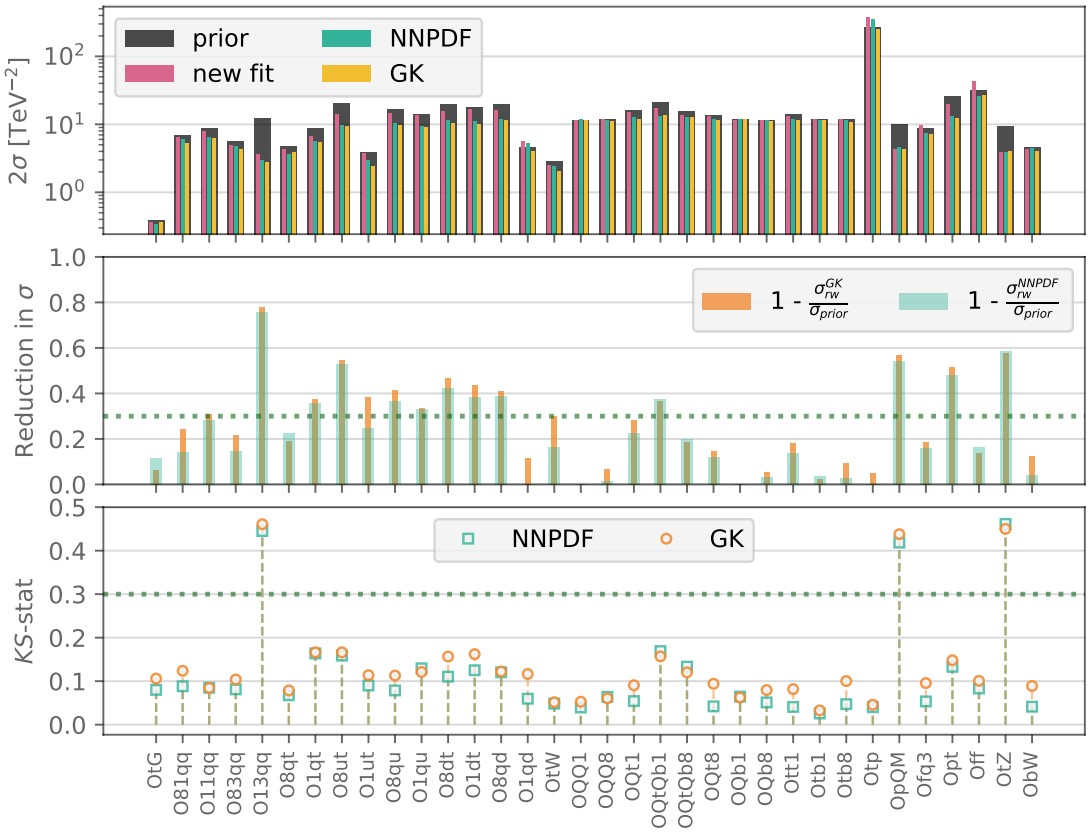

Figure 4.4: Same as Fig. 4.1 now with only one $t$-channel single-top dataset (the differential $y^{t+\bar{t}}$ distributions from CMS at 8 TeV) added via reweighting

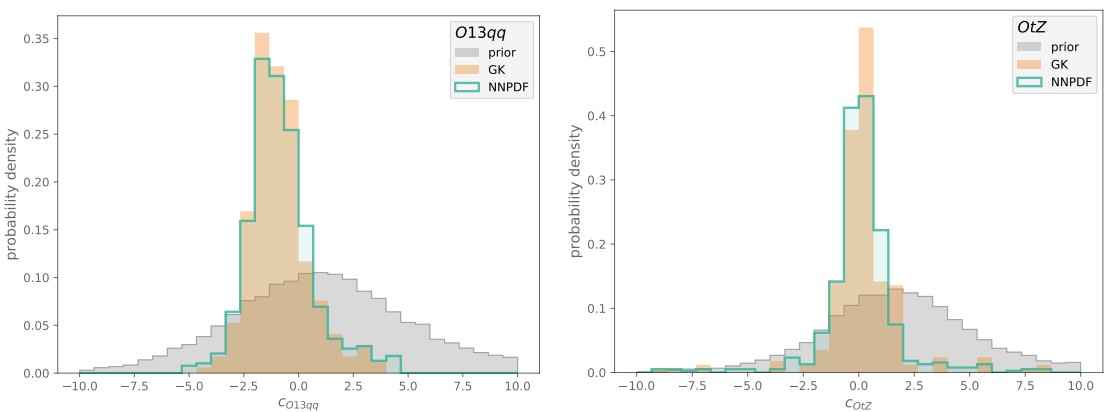

Figure 4.5: Same as Fig. 4.3 now for the case in which only one $t$-channel single-top dataset added via reweighting as opposed to the complete set of $t$-channel measurements, see also the results of Fig. 4.4. The NNPDF results correspond to $N_{\text{eff}} \simeq 600$ while the GK ones to $N_{\text{eff}} \simeq 250$.

superior performance, although we are not aware of any theoretical work supporting such a choice. For completeness, here we investigate the performance of such hybrid weights from the phenomenological point of view.

Specifically, we repeat the reweighting exercise that led to Fig. 3.3, namely including all the $t$-channel single top production measurements. Now, we use a one-parameter family of

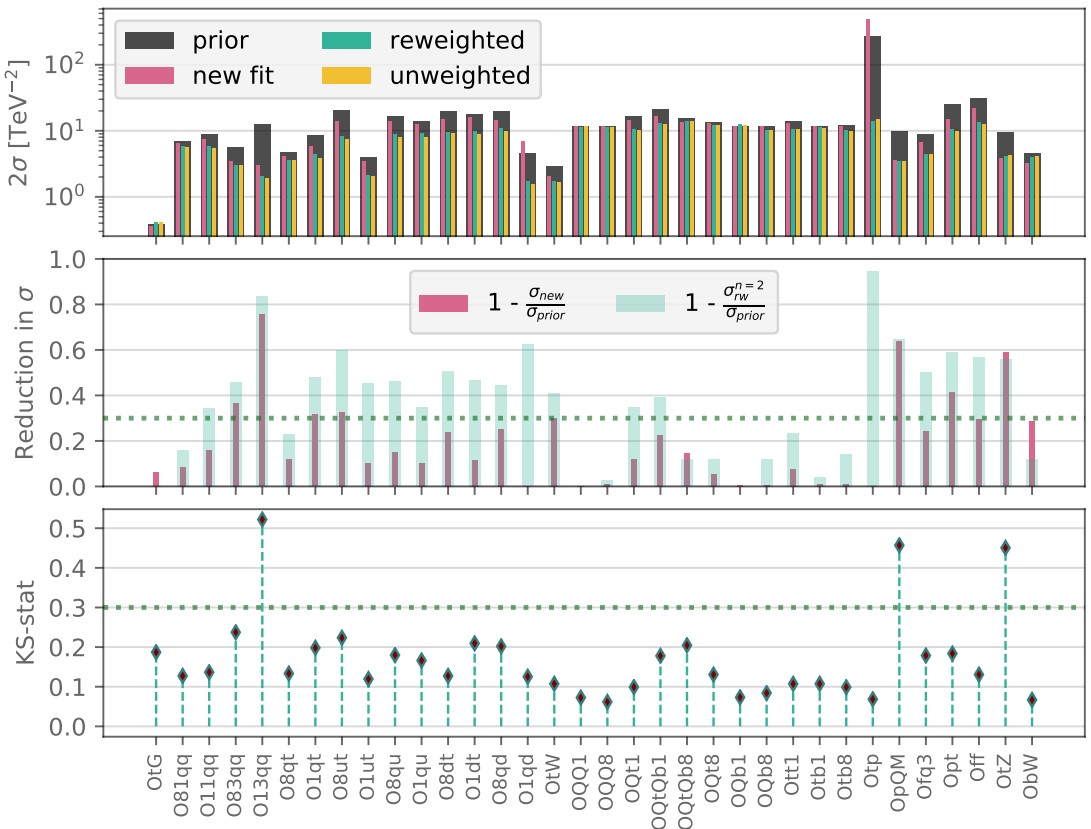

Figure 4.6: Same as Fig. 3.3 but now using the hybrid weights defined in Eq. (4.2) with $p = 2$.

weights

$$\omega_k^{(p)} \propto \left[ \left( \chi_k^2 \right)^{(n_{dat}-1)/2} \right]^{1/p} \exp\left( -\chi_k^2/2 \right), \quad k = 1, \ldots, N_{rep}, \tag{4.2}$$

where $p$ is a parameter that interpolates between the NNPDF weights ($p = 1$), Eq. (2.4), and the GK weights ($p \rightarrow \infty$), Eq. (4.1). We have re-evaluated Figure 3.3 for different choices of $p$, and we have estimated the corresponding effective number of replicas. This is what one finds for two intermediate values, $p = 2$ and $p = 3$:

| $p$ | 1 (NNPDF) | 2 | 3 | $\infty$ (GK) |
|---|---|---|---|---|
| $N_{eff}$ | 306 | 115 | 68 | 22 |

Therefore there is no benefit in using the intermediate weights option: all values of $p$ decrease the efficiency of the reweighting procedure in comparison to the NNPDF case.

Furthermore, we have verified that, provided the effective number of replicas $N_{eff}$ is large enough ($N_{eff} \sim 100$), the results obtained with Eq. (2) still reproduce the corresponding results of a new fit. This is shown explicitly in Fig. 4.6 below, the counterpart of Figure 3.3 but now using the hybrid weights in Eq. (4.2) with $p = 2$. As can be seen, for those operators for which the KS-statistic and the reduction of uncertainties lie above the given threshold, the fit results are well reproduced by the reweighting with these hybrid weights. However, for $p = 2$, we end up with $N_{eff} = 115$ effective replicas, a value significantly smaller than the one obtained in the case of NNPDF weights ($N_{eff} = 306$).

To summarise, we find that the hybrid weights of Eq. (4.2) are equivalent to the NNPDF ones provided that the resulting $N_{\text{eff}}$ is large enough, but also that they are less efficient since $N_{\text{eff}}^{(p>1)} < N_{\text{eff}}^{(p=1)}$. Their use is therefore not advisable.

## 5 Summary

The lack of direct evidence for new physics at the LHC so far makes it crucial to develop indirect pathways to identify possible signatures of BSM dynamics from precision measurements. One of the most powerful frameworks to achieve this goal is provided by the SMEFT, which allows for a theoretically robust interpretation of LHC measurements. Ensuring the model independence of this approach, however, requires us to efficiently explore the large parameter space of Wilson coefficients related to the SMEFT operators. This can be both technically challenging and computationally expensive.

In this work we showed how Bayesian reweighting, an inference method widely used to assess the impact of new data in global determinations of PDFs, can be extended to constrain a Monte Carlo representation of the SMEFT parameter space upon the inclusion of new experimental input. Reweighting consists of assigning a weight to each of the replicas that define the prior probability distribution. These weights are computed as a function of the agreement (or lack thereof) between the prior and the new experimental measurements not included in it. The method has two advantages in comparison to a new fit to an extended set of data: first, it is essentially instantaneous, and second, it can be carried out without needing access to the original SMEFT fitting code. Using single-top production measurements from the LHC, we showed that, under well-defined conditions, the results obtained with reweighting are equivalent to those obtained with a new fit to the extended set of data.

Nevertheless, the results obtained with the reweighting method need to be considered with some care. First, it is necessary to verify that the efficiency loss of the reweighted sample is not so severe as to make the procedure unreliable. In practice, this requires that the effective number of replicas remains sufficiently large. Second, it is necessary to identify those operators for which the results of reweighting are driven by a genuine physical effect, and ignore those affected by large statistical fluctuations or other spurious effects. In practice, this requires that the value of the KS statistic is sufficiently high. We therefore proposed a possible set of selection criteria to identify for which operators the outcome of the reweighting method is expected to coincide with that of a new fit. We quantitatively based our criteria on the values of the KS statistic and of the relative reduction factor for the uncertainties with respect to some threshold.

As a byproduct of this analysis, we also assessed the dependence of our results upon the specific choice of weights, a topic which has been the cause of some discussions within the PDF community in recent years. Specifically, we compared the performance of the NNPDF and GK weights, proposed respectively in [34, 35] and in [36, 37]. We found that, while the two methods lead to comparable results if the new data is not too constraining, the latter is far less efficient than the former, in the sense that the effective number of replicas decreases much faster. Our results provide further evidence in support of the use of the NNPDF rather than the GK expression for applications of Bayesian reweighting to Monte Carlo fits, either in the PDF or in the SMEFT contexts.

The main limitation of the reweighting method is that it requires a large number of starting replicas to be used reliably. The problem is here somewhat more serious than in the PDF case, where an initial sample of $N_{\text{rep}} = 10^3$ replicas is usually sufficient for most practical applications. For instance, a prior of at least $\mathcal{O}\left(10^5\right)$ replicas would be required for a simultaneous reweighting with all the $t$- and $s$-channel single-top data. This happens because, in

the SMEFT case, one is trying to simultaneously constrain a large number of independent directions in the SMEFT parameter space, each corresponding to a different (combination of) operator(s). However the efficiency loss should not represent a limitation to the applicability of the reweighting method, as in practice one usually wants to assess the impact of a single new measurement. Of course, the reweighting method can only be applied to explore directions in the parameter space that are already accessed in the prior set. If new directions are expected to be opened by new data, for example when measurements sensitive to different sectors of the SMEFT are included, reweighting is not applicable and a new fit would need to be carried out.

A `Python` code that implements the reweighting formalism presented in this work and applies it to our SMEFiT analysis of the top quark sector is publicly available from

https://github.com/juanrojochacon/SMEFiT

together with the corresponding user documentation (briefly summarised in the Appendix). In addition to the analysis code, we also make available three prior fits, each of them made of $N_{\rm rep} = 10^4$ Monte Carlo replicas. The first fit does not include any $t$- and $s$-channel single-top quark production measurement, otherwise it is equivalent to the baseline fit of Ref. [23]. It can be used to reproduce the results presented here. The second fit is equivalent to the baseline fit presented in [23], but has $N_{\rm rep} = 10^4$ Monte Carlo replicas instead of $N_{\rm rep} = 10^3$. It can be used to assess how the probability distribution of the top quark sector of the SMEFT is modified by the addition of new measurements via reweighting. The final set is based only on inclusive top-quark pair production measurements.

# Acknowledgments

We are grateful to Stefano Forte for discussions and suggestions related to the results of this paper. S. v. B. is grateful for the hospitality of the Rudolf Peierls Centre for Theoretical Physics at the University of Oxford where part of this work was carried out.

**Funding information**    J. R. and E. S. are supported by the European Research Council Starting Grant "PDF4BSM". J. R. is also partially supported by the Netherlands Organisation for Scientific Research (NWO). E. R. N. is supported by the European Commission through the Marie Skłodowska-Curie Action ParDHonS FFs.TMDs (grant number 752748).

# A    Code documentation

The reweighting code in the publicly available `GitHub` repository consists of a single `Python` script. It can be used straightforwardly by executing `SMEFiT_rw_unw.py` with `Python3`. In order to run the code, the following `Python` packages need to be installed beforehand:

- `numpy`
- `tabulate`
- `scipy`
- `matplotlib`
- `os`

```
 1   '''
 2   +----------------------------------+
 3   | Input settings for reweighting code |
 4   +----------------------------------+
 5   '''
 6
 7   # Set Nrep, KS criterium and required error reduction
 8   n_reps        = 10000
 9   ks_level      = 0.3
10   reduction_level = 0.3
11   produce_plots  = 'on'  # 'on' to produce plots
```

Figure A.1: A snapshot of the python script `code_input.py`. The input settings of the reweighting procedure can be defined here.

**Code input**

Besides the reweighting code file, there is also a second Python file called `code_input.py` that defines the input settings to be used for the reweighting procedure. In this file, the user is able to select the prior SMEFT Monte Carlo analysis, the datasets that he wants to include in it by reweighting, and the number of replicas that should be used from it to this purpose. By modifying this file, the user can also specify the threshold values for the KS statistic and the error reduction factor that determine for which degrees of freedom the reweighted results are reliable. In Fig. A.1, a code snapshot of `code_input.py` is shown.

The following inputs will be required for the reweighting code to be executed:

- The Wilson coefficients that define the prior fit.

  Together with the code, we also release in the `rw_input_data/Wilson_coeffs/` folder the results of three different priors: `all_datasets`, based on the full dataset of [23], `no_single_top`, with single top-quark production data excluded, and `only_ttbar`, based exclusively on top-quark pair-production measurements.

- The replica-by-replica $\chi^2$ computed for the new data using the corresponding theory predictions based on a given prior SMEFT analysis.

  For illustration purposes, here we make available in the `rw_input_data/chi2_data/` folder the files `t_channel`, `1st_t_channel`, and `s_channel`, which are obtained from the prior set `no_single_top` and correspond respectively to all $t$-channel single top-quark data, all $s$-channel single top-quark data, and only the first $t$-channel single top-quark measurement in Table 3.1.

- The Wilson coefficients determined from a new fit to the extended set of data (if available, required only for validation).

  Here, also in the folder `rw_input_data/Wilson_coeffs/`, we provide the results of the `t_channel`, `1st_t_channel`, and `s_channel` fits, which can then be directly compared to the corresponding reweighted results.

**Code work-flow**

The code executes the reweighting and unweighting procedure through the following steps:

```
                         +--------------------------------+
                    ---  | Table of constrained operators |  ---
                         +--------------------------------+

| operator |   prior st dev  |   poster st dev  |  rw st dev  |  unw st dev  |  KS stat |
|----------|-----------------|------------------|-------------|--------------|----------|
| O13qq    |            6.22 |             1.52 |        1.34 |         1.36 |     0.45 |
| OpQM     |            5.01 |             1.80 |        2.05 |         2.07 |     0.46 |
| OtZ      |            4.72 |             1.94 |        1.75 |         1.75 |     0.44 |
```

Figure A.2: A partial snapshot of the code output. The constrained SMEFT operators that satisfy the reweighting selection criteria of Sect. 3 are listed in a table with the value of their corresponding standard deviation and KS statistic.

1. Load in the prior set.
2. Load in the replica-by-replica $\{\chi_k^2\}$ values for the prior predictions.
3. Compute the weights $\omega_k$ for each replica.
4. Determine the Shannon entropy.
5. Obtain the reweighted set and construct the unweighted set.
6. Determine the KS statistic.
7. Load in the new fit for validation (if available).
8. Determine the reduction of the uncertainty for the SMEFT degrees of freedom.
9. Obtain the operators that satisfy the selection criteria defined in Sect. 3.
10. Save results to file and produce validation plots.

**Code output**

When the code is executed from a terminal, its output will be displayed as in Fig. A.2. The following results will be saved in a new folder called `rw_output`:

- The plots of the $2\sigma$ bounds on the reweighted and unweighted Wilson coefficients compared to the prior (and, if available, to the new fit for validation), together with the associated uncertainty reduction factor and the value of the KS statistic.

- The histograms for the distributions of the Wilson coefficients for those operators that satisfy the selection criteria defined in Sect. 3.

- A text file `unw_coeffs.txt` with the results of the unweighted set of Wilson coefficients. In this file, the rows correspond to the replica number in the unweighted set, and the coefficients are separated in columns for each operator.

  Recall that the number of SMEFT operators in this unweighted set will be identical to that of the adopted prior.

Using this code, the results presented in this paper can be easily reproduced by selecting the same input settings as those adopted in the validation exercises presented in Sect. 3.

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
