# Peer review of "Constraining the SMEFT with Bayesian reweighting"

_SciPost Physics, doi:SciPost Phys. 7, 070 (2019)_

## Round 1 · Referee Report · Anonymous (Referee 1) · 2019-7-8

Report

This article is a valuable contribution to the ongoing study of the best way to utilize the SMEFT as a tool to constrain generic new physics using precision measurements at the LHC and other experiments; it explores the possibility of using reweighting techniques to produce a fast approximation to the results of a complete fit in the presence of a new data source.

I note that there is a notational error occurring in equation (2.9) and following; one imagines that $F(_{\rm rw}(\langle c_i\rangle)$ does not need the additional open-parens. On a similar note, there are typos on page 10 (ne for new), in the caption to Figure 4.3 (fo for of), and on page 14 (NNDPF for NNPDF) that jumped out at me.

Moving to address the physics and statistics of the contribution, I do have a couple of questions that aren't fully explained (at least at my level of comprehension) in the article. The most troublesome point is the inclusion of quadratic EFT effects, that are $\mathcal O(\Lambda^{-4})$, in the signal function. The effects at this order in the EFT expansion are not fully given by the terms kept by the authors, and following the usual rules of perturbation theory calculations these ought to be dropped, with a theoretical error introduced to parameterize our ignorance of the size of the effect at this order. With the current treatment, which is sadly common in the field, "constraints" are regularly produced on the EFT parameter space which do not hold in more-complete models, admitting instead model-building workarounds, which would not be the case with a robust theoretical treatment of the EFT expansion. It would at least be very beneficial to understand, in every article produced on this topic, how impactful the "quadratic" terms are on the fit itself (which the authors have tersely explored in their previous work, citation [23]). More generally, it is very important to acknowledge that the EFT is a series expansion in something like $\frac{s}{\Lambda^2}$, and as such it doesn't make sense to consider scales $\Lambda\sim1$ TeV at the LHC; it's perfectly clear that this will not converge. I would recommend then that the authors re-benchmark their scale of new physics to $\Lambda\sim5$ TeV instead.

I'm also confused by the comments regarding double-counting and whether or not it is problematic in this context on page 6; if the goal is to explicitly reproduce the results of the fit of [23] through sub-fitting and then reweighting why should I be including additional data which was excluded there? Does the reweighting procedure meaningfully depend on that data to reproduce the fit accurately? More generally, it isn't clear to me why double-counting would be less of a concern in the reweighting, fast-fit production context than it is in the context of a full-blown fit.

The testing for reliability of reweighted results based on the KS statistic seems to have been employed in a thoroughly ad-hoc way here as well; is there any mathematical/statistical reason why we should expect these thresholds of 0.3 or 0.2 to be dispositive as to the reliability of the reweighting procedure? Given the differences in adopted thresholds for different sub-analyses, is there some meaningful interpolating formula, perhaps one that takes in to account the amount of data points being added, which could suggest a reasonable threshold for reliability in future reweighting exercises? Also, and more worryingly, given that e.g. the result for O13qq is officially reliable but the result for Ofq3, which is very strongly correlated with O13qq in the full fit of [23], is not, how are we to think about correlations and flat directions in the reweighting framework?

I also am struck by the (admittedly not phenomenologically relevant) increasing feature in Figure 4.2 in going from 6 to 7 datasets included; I would naively have assumed that introducing additional data should only be further narrowing the range of replicas that were consistent with the data, but that doesn't seem to be the case here. Is this behavior understood by the authors? A short comment explaining it would be valuable to the reader I believe.

Finally, I find myself thoroughly confused by the discussion of NNPDF versus GK weights; it is clear that having accidentally found something that fits very well can be damaging in the case of GK weights, but it isn't obvious why that sample should be fully discarded as in the NNPDF formalism; fitting well doesn't generally get punished in statistics, but it shouldn't be overly rewarded. Is it plausible that some middle-ground treatment exists, which for instance treats any fit better than that which maximizes the NNPDF weight as equally-worthy with that maximizing fit?

In all this is indeed a valuable exploration of techniques for rapidly estimating the impact of new data on fits in the SMEFT, and deserves to be published after addressing these questions and comments.

Requested changes

  1. Correct typographical errors in notation and text as identified.
  2. Renormalize theory to a cutoff scale $\Lambda\sim5$ TeV where the SMEFT approach is theoretically consistent.
  3. Explore the importance of keeping versus dropping "quadratic" contributions to the reweighting procedure, as well as the impact of introducing new theoretically errors for missing higher orders in $\Lambda^{-2}$.
  4. Explore and discuss the impact of flat directions on reliability metrics for reweighting results.
  5. Add explanation for rising feature to caption of Figure 4.2.
  6. Explore and discuss potential intermediate-case weights for the reweighting procedure.

  • validity: good
  • significance: good
  • originality: high
  • clarity: high
  • formatting: excellent
  • grammar: excellent

Author:  Emanuele Roberto Nocera  on 2019-10-08  [id 626]

(in reply to Report 1 on 2019-07-08)

We thank the referee for her/his comments. Our reply can be found in the attached document.

Attachment:

referee-reply.pdf

---

## Round 1 · Referee Report · Minho Son (Referee 2) · 2019-7-30

Report

In the manuscript, the authors investigate if the Bayesian reweighting can be applicable to the SMEFiT framework (global analysis of SMEFT) using the data set used in previous literature. Performing a new fit on the newly updated data set might be computationally expansive when a large of EFT coefficients and/or a large set of data are involved. The Bayesian reweighting method seems to allow us to estimate the impact of those new extended data on the EFT parameter space.

I find that the approach proposed in the manuscript is an interesting idea and it will be eventually very useful if it works as was described. The importance of the global fit in the SMEFT is growing, and we see various examples where the global fit makes a significant difference. However, even at the level of the dimension-six operators it is difficult to carry out the global fit due to a large number of operators. This type of approach in the manuscript looks in the right direction to make the SMEFT more practical and to improve the mapping between data and Wilson Coefficients (WCs) in a more accurate way.

I feel that the review in Section 2 is too short to fully understand how the method works. I would suggest to make it more self-contained. The remaining of the manuscript is clearly written and organized.

I only have a few basic questions. I do not mean that they have to be implemented in the revision if they are too basic or not necessary. It is up to the authors.

  1. Section 2: It would be helpful if some explanation on how $N_{rep}$ Monte Carlo replicas are generated from the initial data set (instead of just giving a ref) since it seems essential to understand it. Similarly for the chi-squared in eq. 2.4. When comparing the theory prediction from each replica with the new data set in chi-squared, is the uncertainty of the new data set also taken into account?

  2. Section 2: Is it obvious why all $N_{op}$ WCs in $k$-th replica has a universal weight? A theory prediction (for a new measurement) constructed from $k$-th replca might involve smaller set of WCs. Then, do not we need reweighting of those finite subset of WCs?

  3. Section 2: In this approach, is it essential to have an ensemble of all WCs (including those that will be covered in the new data set) in the prior irrespective of the status of the initial data set? For instance, suppose, we have an initial data set that is sensitive to a subset of WCs (say set I). Later, new data set is added, and new one is sensitive to mostly different types of WCs (say set II) compared to the initial ones. Is it correct that at least sets I $\cup$ II have to be included in the ensemble of prior for this approach to work? In this situation, if we decide to add a second new data set later which is sensitive to WCs set III. Then, we have to go back to the beginning and should start again with constructing an ensemble of MC replicas for WCs in sets I $\cup$ II $\cup$ III?

  4. Section 3.2: In Fig. 3.2, $N_{eff}$ abruptly drops when the $s-$channel measurements are subsequently added. It is explained that it is due to a large amount of new information being added, specifically sensitivity to new combinations of SMEFT parameters that are unconstrained by the measurements previously considered. I feel that this goes against what one does in the SMEFT. In the New Physics search via the SMEFT, our goal would be to come up with as many new measurements that give constraints on new combinations of WCs as possible or constraints on as many new set of WCs as possible. It sounds like the better new types of processes we come up with, the less the approach become effective. I wonder if the authors have any comment on this.

I strongly recommend the current work to be considered for the publication in SciPost Physics after the questions above are clarified.

Requested changes

  1. Please improve section 2 so that it becomes more self-contained.

  • validity: good
  • significance: high
  • originality: high
  • clarity: good
  • formatting: excellent
  • grammar: excellent

Author:  Emanuele Roberto Nocera  on 2019-10-02  [id 619]

(in reply to Report 2 by Minho Son on 2019-07-30)

We thank the referee for her/his detailed comments, which we address in the attached document.

Attachment:

referee-reply.pdf

---

## Round 2 · Referee Report · Anonymous · 2019-11-1

Report

I am very satisfied by the edits made in response to previous refereeing, and recommend this article for publication.

---

## Round 2 · Referee Report · Anonymous · 2019-11-19

Report

All issues were clearly clarified by the authors. I strongly recommend the manuscript for the publication in the SciPost.

---

## Round 2 · Author Response

See attachment.

---

## Editorial Decision

published